# Spatial Distribution Patterns and Influencing Factors of Dominant Species in Plain Valley Forests of the Irtysh River Basin

Jihu Song [1,2,†], Zhifang Xue [1,2,†], Bin Yang [1,2,†], Tong Liu [1,2,*], Ye Yuan [1,2], Ling Xu [1,2] and Zidong Zhang [1,2]

1   College of Life Science, Shihezi University, Shihezi 832003, China; songjihu@stu.shzu.edu.cn (J.S.); xuezhifang@stu.shzu.edu.cn (Z.X.); yb940195356@gmail.com (B.Y.); yuanye@stu.shzu.edu.cn (Y.Y.); xuling@stu.shzu.edu.cn (L.X.); zhangzidong@stu.shzu.edu.cn (Z.Z.)
2   Xinjiang Production and Construction Corps Key Laboratory of Oasis Town and Mountain-Basin System Ecology, Shihezi 832003, China
*   Correspondence: liutong@shzu.edu.cn; Tel.: +86-0993-13579751189
†   These authors contributed equally to this work.

**Abstract:** The Irtysh River, which stretches for 633 km, is the second longest river in Xinjiang. The valley forests within its basin are unique forest resources that exhibit crucial ecological functions and form an integral part of China's "Three North" Shelterbelt Forest Project. However, previous studies mainly focused on individual tributaries or main streams, lacking comprehensive research on the overall river and valley forest resources and their ecological functions. To address this research gap based on comprehensive investigations, this study analyzed the dominant species composition, spatial distribution patterns, and influencing factors of valley forests across various branches of the Irtysh River basin plain. The results revealed the presence of 10 local tree species in the area, with *Populus laurifolia*, *Populus alba*, *Salix alba*, and *Betula pendula* as the dominant species. However, seedling regeneration was relatively weak. *P. laurifolia*, *P. alba*, and *S. alba* were widely distributed across tributaries and main streams, whereas *B. pendula* was primarily found in the tributaries. The four dominant species exhibited distinct clustering patterns. The concentration intensity of these dominant species in the main stream of the Irtysh River basin was significantly higher than those in other tributaries, with *P. laurifolia* showing a lower concentration intensity across the entire basin than the other dominant species. Negative density dependence was the primary biological factor influencing species aggregation intensity, with significant positive effects on *P. alba* and *S. alba* and significant negative effects on *B. pendula*. Among the abiotic factors, elevation had a significant positive effect on the aggregation intensities of *P. alba*, *S. alba*, and *B. pendula*, indicating that these species tend to aggregate more densely at higher elevations. Conversely, slope had a significant negative impact on the aggregation intensities of *P. laurifolia*, *P. alba*, and *S. alba*, suggesting that increasing slope steepness leads to a decrease in the clustering of these species. Similarly, the distance from the river channel had a significant negative effect on the aggregation intensities of *S. alba* and *B. pendula*, implying that as the distance from the river increases, the clustering patterns of these species become less pronounced. This study aimed to detail the current state of valley forest resources and their ecological functions, thereby laying a foundation for their effective protection.

**Keywords:** population distribution pattern; aggregation intensity; negative density dependence; size structure; plain valley forests

## 1. Introduction

Valley forests, which develop along the floodplains and coasts of periodically flooded lowland valley zones [1], are widely distributed in subtropical and temperate regions [2–5]. These forests perform crucial ecosystem functions and services and facilitate a high ecosystem productivity [6], and valley forests accumulate carbon faster than other dryland forests,

contributing significantly to rapid carbon sequestration [7]. Valley forests provide critical habitats for plants and animals [8,9] and help reduce riverbank erosion caused by water flows [10]. Additionally, these forests act as buffers between water flows and agricultural landscapes, mitigating flood disturbances to cash crops, protecting water quality, and regulating microclimates and water temperatures [11,12]. Furthermore, they are essential in maintaining ecosystem integrity and mitigating climate change [13].

The spatial pattern of plant populations refers to their typical spatial distribution structure, which depends on the ecological niches of plant resources, plant competition, and environmental adaptability [14]. In a given environment, the effective use of resources and space by plant populations contributes to an optimal spatial layout [15]. This pattern is influenced by the biological characteristics of species and competition among populations, as well as by habitat factors, including soil, topography, and geomorphic conditions [16,17]. Growth, reproduction, resource utilization, and competition for dominant species directly or indirectly affect the formation and maintenance of a community's structure and succession [18,19]. Therefore, studying the spatial distribution patterns of dominant species populations is essential for analyzing the adaptation status and ecological functions of community species, as well as for implementing effective conservation measures.

The Irtysh River originates from the southern foot of the Altai Mountains in northern Xinjiang, China, flows through China, Kazakhstan, and Russia, and eventually empties into the Arctic Ocean. As the river traverses the northwest portion of the Junggar Basin, it supports a complex ecosystem characterized by both mountainous cold temperate coniferous forests and arid valley vegetation, thus demonstrating its crucial ecological function. The valley forest is rich in rare endemic tree species, particularly within the Salicaceae and Betulaceae families, making it a key area for the global diversity of Salicaceae plants [20,21]. However, human activities, such as water conservation and hydropower projects in the Irtysh River Basin, have severely disturbed the river's hydrological conditions and significantly impacted the habitats of both animals and plants [22]. Therefore, the effects of these disturbances on the survival and development of valley forests must be thoroughly evaluated. Although some scholars have studied poplar genetic diversity [23,24], ecological water demand and dynamics of forests and grasslands in valleys [25], ecosystem assessments [26,27], plant diversity [28], and vegetation changes [29], a significant limitation of these studies lies in their narrow focus. Specifically, most prior works have been constrained to individual tributaries or the main streams within the study basin, failing to capture the broader context and interconnectedness of valley forests across the entire valley system. Furthermore, little attention has been paid to the distribution and ecological roles of dominant species within these forests, which are vital for understanding their resilience and sustainability. Therefore, there is a pressing need for a comprehensive documentation of the current state of river and valley forest resources, including a detailed assessment of their ecological functions and the distribution of their dominant species. Such an endeavor would fill the gaps in our knowledge and provide a solid foundation for the effective conservation, management, and restoration of these valuable ecosystems.

Based on the topography of the valley in the study area, valley forests are classified into plain and mountain valley forests. The plain valley forest is primarily located in low-elevation plain valley areas with a relatively flat terrain, comprising four-fifths of the total valley forest area in the Irtysh River basin. This area serves as the main ecological functional zone and is the most affected by human activities. Accordingly, this study focused on the plain valley forest in the Irtysh River basin. Through a comprehensive survey of the main stream of the Irtysh River as well as the tributaries of the Crane, Burqin, Haba, and Berezek rivers, this study sought to detail the following three aspects: (1). Identification of the dominant species in the plain valley forests of each branch and their regeneration. (2). Define the spatial distribution patterns of the dominant species in the valley forests of the main stream plains of each branch. (3). Determine the effects of biological (negative density dependence) and abiotic factors (elevation, slope, and distance from the river channel) on the spatial distribution patterns of the dominant species. This study sought to

elucidate the growth and distribution status of valley forests and provide data to support the protection, rational development, and utilization of local valley forests.

## 2. Materials and Methods

### 2.1. Study Sites

The study area is located in the Irtysh River basin (46°00′–49°10′ N, 85°31′–91°04′ E) in Altay Prefecture, Xinjiang, China (Figure 1). Originating at the southern foot of the Altai Mountains, the Irtysh River covers an area of 57,000 km² in China and extends 633 km. The basin forms a typical river system owing to its topography and geomorphological landscape. The Irtysh River basin encompasses the main stream of the Irtysh River and its primary tributaries, including the Kara Irtysh, Crane, Burqin, Haba, and Berezek rivers. Each of these rivers supports a unique distribution of valley forests along their respective valleys, contributing to the diverse and rich ecosystem of the entire basin.

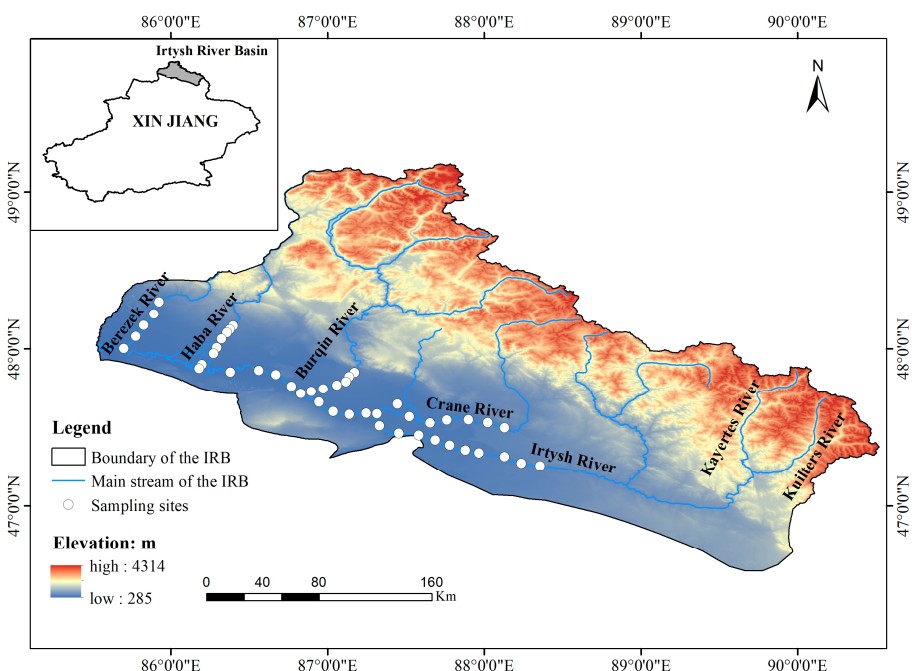

**Figure 1.** Location of the study area and distribution of sample plots (plains areas without sample plots did not have valley forests or had sparse trees that did not meet the conditions for installing sample plots).

The front plain of Altai, through which the Irtysh River flows, is located at the northern margin of Junggar Basin. The terrain slopes from east to west with an elevation range of 300–600 m. This area, which is far from the sea and influenced by the surrounding alpine basins, exhibits typical arid climatic characteristics. The average annual rainfall is less than 120 mm, whereas the average annual evaporation exceeds 1500 mm. The average annual temperature is 4.2 °C, with the average annual minimum temperature reaching −28.1 °C (Table 1).

**Table 1.** Overview of the main and tributary streams in the Irtysh River basin.

| River Name | Type of Water Body | Total Length (km) | Drainage Area (km²) | Distance from Mountain Pass (km) |
|---|---|---|---|---|
| Irtysh River | Main stream | 633 | 57,000 | 256 |
| Crane River | Primary tributary | 265 | 6792 | 74 |
| Burqin River | Primary tributary | 269.6 | 9960 | 42 |
| Haba River | Primary tributary | 216.3 | 7224 | 40 |
| Berezek River | Primary tributary | 155 | 1600 | 54 |

## 2.2. Data Collection

Based on a comprehensive field survey conducted during the growth period of the valley forest, plots were distributed according to gradient changes in the dominant species in the vegetation communities. These plots were located along the main stream of the Irtysh River and the tributaries of the Crane, Burqin, Haba, and Berezek rivers (Figure 1). Sample plots were established every 4–8 km from the mountain pass of each tributary to the junction with the main stream. Along the main stream, plots were established every 8–12 km. In total, 48 sample plots were installed: 20 in the main stream, 8 in the Crane River, 7 in the Burqin River, 8 in the Haba River, and 5 in the Berezek River.

Based on the community structure and species composition, four quadrats were established in each plot, varying by distance from the river channel. Each quadrat, measuring 30 m × 30 m, was spaced at intervals of no less than 30 m, resulting in 192 quadrats. Each 30 m × 30 m quadrat was further divided into nine 10 m × 10 m quadrats using the adjacent grid method. A GARMIN (Garmin Corp, New Taipei City, Taiwan, China) 631csx GPS was used to record the latitude and longitude of each plot. The species name, DBH (diameter at breast height), tree height, crown width, and number of tree layers (for trees with a diameter ≥3 cm) were recorded for each tree quadrat. Using ALOS PALSAR DEM data with a 12.5 m accuracy provided by ASF DAAC (https://search.asf.alaska.edu/#/, accessed on 19 November 2023), terrain factors such as elevation, slope, and distance from the river channel were extracted through base terrain analysis using the ArcGIS Pro 3.0.2 software (ESRI, Redlands, CA, USA).

Based on references such as "Flora of China", "A Brief Flora of Xinjiang", iPlant Flora (https://www.iplant.cn/, accessed on 19 November 2023), the Chinese Virtual Herbarium (CVH, https://www.cvh.ac.cn/, accessed on 19 November 2023), the National Specimen Information Infrastructure (NSII), and the Chinese Plant Species Information Database (https://www.plantplus.cn/, accessed on 19 November 2023), we have accurately identified the plants and supplemented species information, including the scientific name of the family, genus, and species.

## 2.3. Identification of Dominant Species and Diameter Class Structure

For trees, important values were calculated using the relative abundance, relative frequency, and relative dominance. The dominant tree species were identified based on these important values, allowing determining the distribution and ecological function of different species in the valley forests of the main stream plain of the Irtysh River basin.

Tree importance value = (relative dominance + relative frequency + relative abundance)/3, and tree dominance was calculated using the chest-height area. Relative dominance = (sum of breast height area of a species/sum of breast height area of all species) × 100, relative frequency = (sum of frequency of a species/sum of frequency of all species) × 100, and relative abundance = (number of species/sum of number of all species) × 100.

DBH was used to estimate the age structure of the standing trees. As multiple species were evaluated, no specific grades were set for DBH. Instead, the number of standing trees within 10 cm DBH intervals was counted to reflect the age distribution and quantity of dominant species in the valley forests of each branch plain.

## 2.4. Analysis of Spatial Distribution Patterns and Population Aggregation Intensity

Aggregation intensity is a key index for evaluating the concentration of population distribution patterns. This index can reveal changes in the aggregation characteristics of the same population over time or under different habitat conditions. Additionally, aggregation intensity allows for simultaneous comparisons of the aggregation status of different populations in similar habitats. Various aggregation indexes offer multiple methods for measuring the aggregation intensity of a single population and provide a multidimensional perspective, thereby enhancing our understanding of population aggregation characteristics.

In this paper, we selected 6 parameters, including the negative binomial parameter ($K$), Cassie index ($C_A$), clumping indicator ($I$), average crowding degree ($m^*$), cluster indicators ($PI$), and diffusion index ($I_\delta$), to judge the aggregation intensity of a population [30,31].

(1) The negative binomial parameter can be defined as follows:

$$K = \overline{X}^2 / (S^2 - \overline{X}) \tag{1}$$

In the above equation, the $K$ value is suitable for measuring the degree of aggregation; the smaller the value of $K$, the higher the degree of aggregation.

(2) The Cassie index is defined as follows:

$$C_A = (S^2 - \overline{X}) / \overline{X}^2 \tag{2}$$

$C_A < 0$ indicates a uniform distribution, $C_A = 0$ indicates a random distribution, and $C_A > 0$ indicates an aggregated distribution.

(3) The clumping indicator is defined as follows:

$$I = S^2 / \overline{X} - 1 \tag{3}$$

$I < 0$ indicates a uniform distribution, $I = 0$ indicates a random distribution, and $I > 0$ indicates an aggregated distribution.

(4) The average degree of crowding is defined as follows:

$$m^* = \overline{X} + S^2 / (\overline{X} - 1) \tag{4}$$

$m^*$ is the average number of neighbors for biological individuals in a quadrat, where the larger the value of $m^*$, the higher the degree of aggregation.

(5) Cluster indicators are defined as follows:

$$PI = m^* / \overline{X} \tag{5}$$

$PI < 1$ indicates a uniform distribution, $PI = 1$ indicates a random distribution, and $PI > 1$ indicates an aggregated distribution.

(6) Diffusion indicators are defined as follows:

$$I_\delta = n(\sum X_i^2 - N) / [N(N - 1)] \tag{6}$$

$I_\delta < 1$ indicates a uniform distribution, $I_\delta = 1$ shows a random distribution, and $I_\delta > 1$ represents an aggregated distribution.

In Equations (1)–(6), population density was used as a quantitative trait for measurement, where $\overline{X}$ is the mean of the various population abundances, $S^2$ is the variance in the various population abundances, $X_i$ is the number of plants in the ith quadrat, $n$ is the total number of plots, and $N$ is the total number of plants in all the plots.

### 2.5. Analysis of Biological Factors in Population Distribution Patterns

Negative density dependence regulates the population density by increasing the mortality rate when the population density becomes too high, leading to a specific spatial distribution pattern. This effect, influenced by natural selection and environmental pressure, helps to maintain a relatively balanced population density, thus preventing excessive resource consumption and uncontrolled population expansion. Therefore, we studied the effects of conspecific and heterospecific negative density dependence to determine their biological influence on population distribution.

Trees were classified as either adults or juveniles based on their diameter at breast height (DBH), measured 1.3 m above the ground. Although tree size is not an exact indicator of age, larger trees are generally older than smaller ones.

Within each species, trees were ranked by DBH and the 99th percentile (dbh99) was determined. All trees with DBH $\geq$ dbh99$^{2/3}$ were classed as adults, whereas trees with DBH < dbh99$^{1/2}$ were classed as juveniles (trees with sizes of dbh99$^{1/2}$ $\leq$ DBH < dbh99$^{2/3}$ were excluded to accentuate the difference between classes) [32,33]; "dbh99$^{2/3}$" refers to the power of two-thirds of the 99th percentile DBH, and "dbh99$^{1/2}$" refers to the power of half of the 99th percentile DBH. The 99th percentile was used to mitigate the influence of outliers on the classification of adult and juvenile trees, ensuring that larger trees were included in the adult category for subsequent analyses.

Using the model developed by Johnson et al. [34], we derived a single parameter to represent the intensity of the density dependence. The glm.nb function in the MASS package was used to fit the negative exponential function to the relationship between the number of saplings and adult trees using the maximum-likelihood method. Specifically, the equation has the form $S = a \cdot e^{bT}$, where S is the number of saplings, T is the number of adult trees, and a and b are parameters fitted using a negative binomial distribution. In this context, a represents the Y-axis intercept and b determines the inflection point of the curve. This equation provides a single parameter, b, to represent the strength of density dependence. The greater the absolute value of b, the stronger the negative density dependence.

We tested for differences in the distributions of the estimates for the strength of conspecific and heterospecific density dependence using the non-parametric Wilcoxon rank-sum test (also known as the Mann–Whitney U test) using the wilcox.test function (Figure 2).

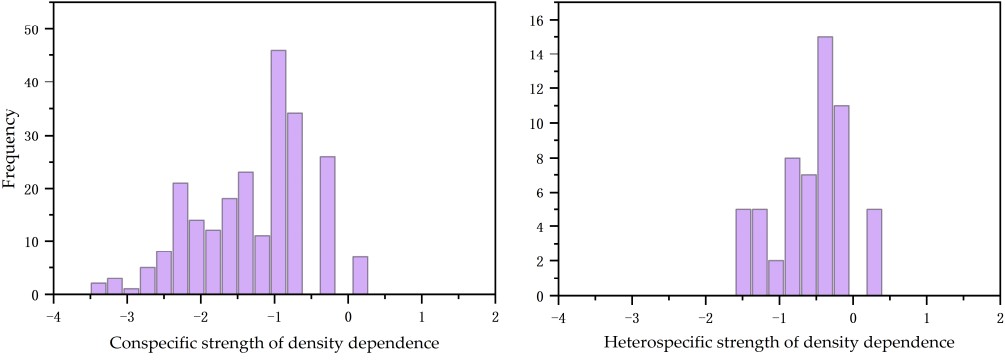

**Figure 2.** Conspecific and heterospecific strength of density dependence.

An analysis of differences in the distributions of allogenic and heterogenic density-dependent intensity estimates showed that populations were more likely to be negatively affected by allogenic influences than heterogenic influences during the establishment phase (Wilcoxon rank sum test: $W = 3352$, $p < 3.79 \times 10^{-9}$). Consequently, the influence of negative density dependence on aggregation intensity was analyzed using the conspecific negative density dependence.

### 2.6. Analysis of Abiotic Factors in Population Distribution Patterns

Abiotic factors such as climate, topography, soil conditions, hydrological characteristics, and natural disasters collectively influence the distribution and growth of plants. However, this study primarily focuses on the impact of topography on the spatial distribution of dominant species in river valley forests. Topographical features, including elevation, slope, and distance from the river channel, affect the water and nutrient availability for plant growth.

Soil is a key factor in determining plant distribution. However, because the trees in this study were primarily distributed and grew on the floodplain of the river alluvium, the soil was mainly sandy. The physical and chemical properties of the soil were relatively consistent, except for the water content. Therefore, a statistical analysis of the soil properties was not conducted in this study.

The Cassie index was used to quantify the aggregation intensity, and a structural equation model was employed to fit the relationships between the negative density dependence, elevation, slope, distance from the river channel, and aggregation intensity of the dominant species in the valley forests of each branch stream plain.

Using a variance decomposition analysis, we determined the contributions of the negative density dependence, elevation, slope, and distance from the river to the aggregation intensity of dominant species in the valley forests of the main stream plain of the Irtysh River basin. This analysis identified the contributing factors and quantified their respective degrees of contribution.

### 2.7. Data Processing and Analysis

R version 4.3.3 was used to perform the study's analyses, with the MASS and stats packages employed for negative density dependence intensity analysis, the lavaan package for structural equation model construction, and the vegan package for the variance decomposition analysis. Data were collated using Excel 2021, and Origin 2021 and ArcGIS Pro 3.0.2 software were used for plotting.

## 3. Results

### 3.1. Species Composition and Regeneration of Trees in the Plain Valley Forest

An analysis of the tree species composition and important values in the plain valley forests of various tributaries and the main stream of the Irtysh River basin revealed that there are a total of 10 tree species, among which *P. laurifolia*, *P. alba*, *S. alba*, and *B. pendula* emerged as the dominant species in the region.

Based on the diameter class structure of the dominant species in each branch stream (Figure 3), there were few young and new seedlings, and a high proportion of middle-aged and mature individuals, indicating a weak population renewal. The dominant species in each main stream were most abundant in the DBH range of 20–60 cm, reflecting a lack of regenerative ability. The DBH of trees in the main streams of the Irtysh River and Crane River Valley forest communities was larger than those of trees in the other three tributaries. This difference was partly due to the variance in dominant species, as *S. alba* and *P. alba* had larger DBHs than *P. laurifolia* and *B. pendula*. *P. laurifolia* was present in the valley forest communities of the Irtysh, Burqin, and Haba River plains but exhibited a wider DBH range in the Irtysh River (Table 2).

**Table 2.** Composition and importance values of tree species in the plain valley forests of the Irtysh River basin.

| Species | Main Stream | | Primary Tributary | | |
|---|---|---|---|---|---|
| | **Irtysh River** | **Crane River** | **Burqin River** | **Haba River** | **Berezek River** |
| *Populus laurifolia* | 31.67 | 19.42 | 57.29 | 22.64 | 7.45 |
| *Betula pendula* | / | / | 39.54 | 58.12 | 12.21 |
| *Salix alba* | 27.78 | 45.88 | 2.33 | 8.85 | 1.33 |
| *Populus nigra* | 4.87 | <1.00 | / | 3.91 | 1.00 |
| *Populus jrtyschensis* | 3.01 | 2.52 | <1.00 | 1.54 | <1.00 |
| *Populus alba* | 29.33 | 31.38 | / | 4.94 | 70.18 |
| *Populus canescens* | / | / | / | / | 7.49 |
| *Populus euphratica* | 3.33 | / | / | / | / |
| *Elaeagnus angustifolia* | / | <1.00 | / | / | / |
| *Salix burqinensis* | / | / | <1.00 | / | / |

Note: "/" indicates that the species does not exist.

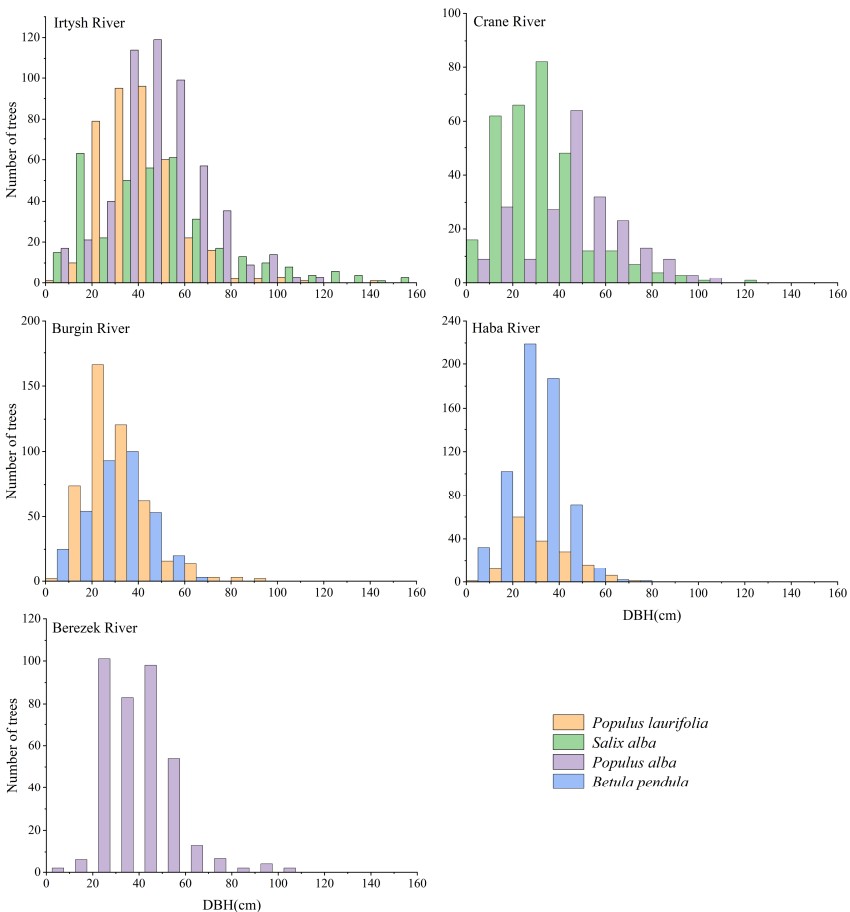

**Figure 3.** Diameter class structure of dominant species in the plain valley forests of each main and tributary stream.

### 3.2. Spatial Distribution Pattern of Dominant Species in Plain Valley Forests

In this study, six parameters were used to determine the distribution patterns of the dominant species in the valley forests of each branch stream plain. The results indicated that the negative binomial parameters, Cassie index, and clumping indicators of the dominant species in each branch stream were greater than 0. Additionally, the average degrees of crowding, clustering indicators, and diffusion indicators were all greater than 1. These findings suggested that the dominant species in the valley forests of each branch stream plain were clustered (Figure 4).

In the plain valley forests along the Irtysh River basin, significant differences in aggregation intensity were observed among the *P. laurifolia*, *P. alba*, *S. alba*, and *B. pendula* populations in various tributaries and main stream plain valley forests, indicating their distinct biological characteristics and varying abilities to adapt to the environment. The lower aggregation intensity of *P. laurifolia*, *P. alba*, and *S. alba* in the main stream of the Irtysh River compared to its tributaries suggests that the main stream environment is more conducive to their growth (Figure 5).

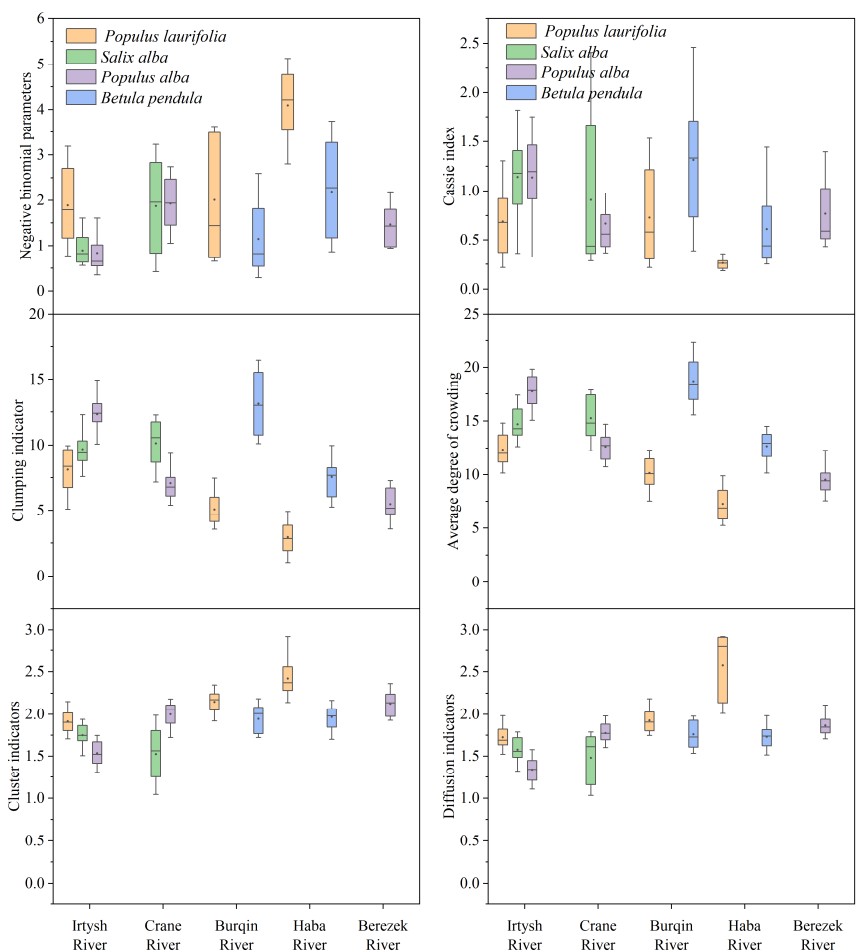

**Figure 4.** Aggregation parameters of dominant species in the plain valley forests of each main and tributary stream.

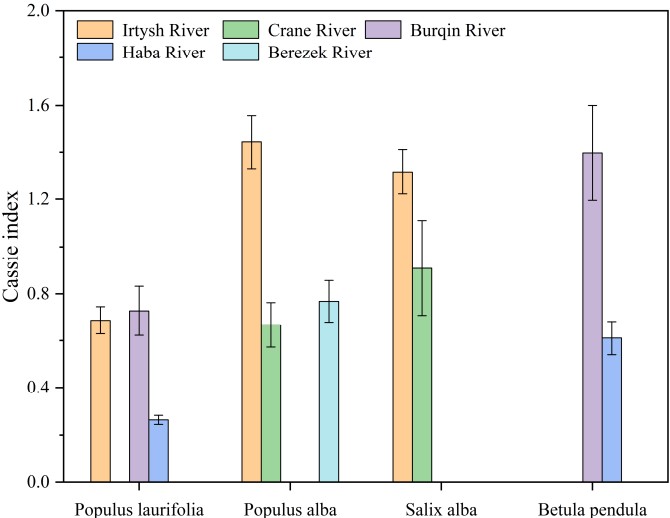

**Figure 5.** Differences in aggregation intensity of dominant species in the plain valley forests of each main and tributary stream.

### 3.3. Factors Influencing the Spatial Distribution Pattern of Dominant Species in Plain Valley Forests

Negative density dependence was the primary biological factor influencing the species aggregation intensity. This factor had a significant positive effect on *P. alba* and *S. alba*

and a significant negative effect on *B. pendula*. Among the abiotic factors, elevation had a significant positive effect on the aggregation intensities of *P. alba*, *S. alba*, and *B. pendula*. Slope had a significant negative effect on the aggregation intensities of *P. laurifolia*, *P. alba*, and *S. alba*. The distance from the river channel had a significant negative effect on the aggregation intensity of *S. alba* and *B. pendula* (Figure 6).

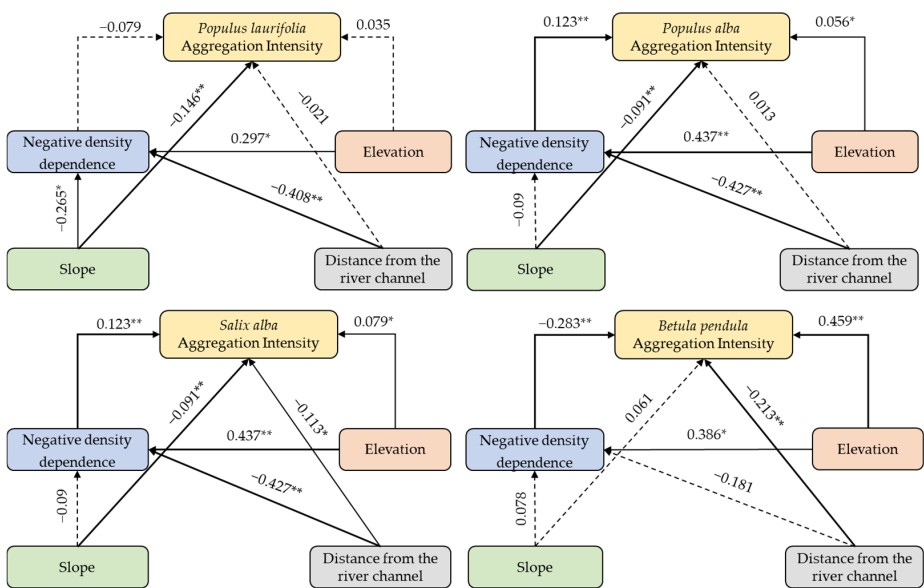

**Figure 6.** Structural equation model of aggregation intensity and influencing factors for different dominant species. Dashed lines indicate that the path has no significant effect (n; *p* > 0.05), the thin line indicates that the path plays a significant role (*; *p* < 0.05), and the thick line indicates that the path plays a very significant role (**; *p* < 0.01).

The variance decomposition analysis of the aggregation intensity and the factors of negative density dependence, elevation, slope, and distance from the river channel for dominant species in the main stream plain of the Irtysh River basin (Figure 7) showed that negative density dependence had the greatest effect on *P. laurifolia*, *P. alba*, and *B. pendula*, whereas distance from the river channel had the greatest effect on *S. alba*. The total explanatory power of these four factors was 59, 64, 79, and 72% for *P. laurifolia*, *P. alba*, *S. alba*, and *B. pendula*, respectively, with *S. alba* having the highest explanatory value.

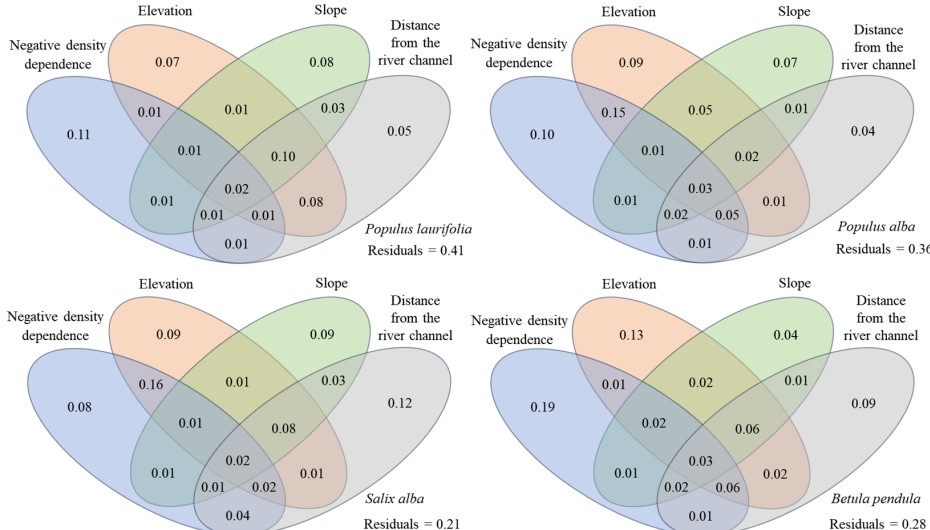

**Figure 7.** Variance decomposition analysis of factors influencing the aggregation intensity of dominant species in the plain valley forests of each main and tributary stream.

In conclusion, negative density dependence, elevation, slope, and distance from the river channel significantly influenced the distribution patterns of dominant species in valley forests. However, these effects varied among species, and the magnitudes of these impacts were inconsistent.

## 4. Discussion

### 4.1. Dominant Species in the Plain Valley Forests of the Irtysh River Basin and Population Regeneration Implications

*P. laurifolia*, *P. alba*, and *Populus jrtyschensis* are endemic species that are naturally distributed in the Irtysh River basin. In the valley forests of the main stream plain, *P. laurifolia*, *P. alba*, and *S. alba* were the dominant species. The dominant species in the valley forests of the tributary plains were *B. pendula*, *P. laurifolia*, *S. alba*, and *P. alba*. Other species of the Salicaceae family, such as *Populus nigra*, *Populus canescens*, *Populus euphratica*, and *Salix burqinensis*, were present in smaller numbers and distributed in a scattered and intermixed manner within the valley forest community.

A comparison of studies on valley forests from different regions worldwide has revealed that forests dominated by Salicaceae species do not exhibit clear zonal patterns and are mainly influenced by local environmental factors. However, a common successional trend was evident: Salicaceae species initially formed pioneer communities. Over time, other deciduous broad-leaved species integrated into these communities and eventually dominated them.

For instance, studies of valley forests along the Mississippi and Wisconsin rivers have shown the dominance of species such as *Acer saccharinum*, *Fraxinus pennsylvanica*, and *Ulmus americana* [4,35]. The dominant species in the Colorado River Valley are *Ulmus crassifolia* and *Fraxinus pennsylvanica* [36]. In Europe, the Upper Rhine River Valley forests are primarily dominated by *S. alba* and *Populus nigra* [4], whereas in the Western Carpathians, *Alnus incana* and *Salix fragilis* are the dominant species [37]. In warmer climates, owing to the higher diversity of tree species, successional stages are longer and more complex, spanning 50 to 200 years [4].

The present study found that in the plain valley forests of the Irtysh River basin, the regeneration of the dominant species was weak. In using diameter classes instead of age to analyze the age structure of the valley forest, there were more large-diameter individuals and fewer saplings and young trees. Dividing the diameter classes into DBH $\leq$ 10 cm, 10 < DBH $\leq$ 30 cm, and DBH > 30 cm, the proportions for *P. laurifolia* were 0.4, 39.6, and 60.0%, respectively; for *P. alba*, the proportions were 2.5, 18.3, and 79.2%. For *S. alba*, the respective proportions were 4.6, 31.4, and 64.0%, and for *B. pendula*, the proportions were 5.8%, 48.0%, and 46.2%.

### 4.2. Clustered Spatial Distribution Pattern of Dominant Species in the Plain Valley Forest

The spatial distribution of plants primarily falls into three categories: clustered, random, and uniform. Because individuals of the same species generally have similar environmental requirements [38], plant populations typically exhibit clustered distributions under natural conditions [39].

This study found that the dominant species *P. laurifolia*, *P. alba*, *S. alba*, and *B. pendula* exhibited a clustered spatial distribution. This pattern was a result of various ecological and biological factors. First, in terms of water resource utilization, valley forests grow in floodplain areas, but the size of the floodplain and thickness of the soil layer determine the preferences of different species. The investigation revealed that species such as *S. alba*, which are tolerant to wet conditions, tend to grow closer to the river to uptake abundant water and nutrients. In contrast, species such as *P. alba* can grow in slightly drier areas. Therefore, the relative uniformity of local floodplain habitats and large-scale spatial heterogeneity were the main factors contributing to the clustered distribution of valley forests in the Irtysh River basin.

Second, although *P. laurifolia*, *P. alba*, *S. alba*, and *B. pendula* can reproduce sexually, long-term grass cutting and grazing by local herders hinder seedling establishment and growth. Consequently, these trees commonly rely on root suckering for reproduction, leading to a clustered distribution of the same species near mature plants.

*4.3. Influence of Negative Density Dependence, Elevation, Slope, and Distance from the River Channel on the Aggregation Intensity of Each Dominant Species*

The spatial distribution patterns of plant populations are related to the biological characteristics of the species and intra- and interspecific competition, as well as those closely associated with their habitats, including topographical and geomorphological conditions [16,17].

Negative density dependence, as a manifestation of population competition, plays a crucial role in shaping the spatial distribution of plant populations [40]. This phenomenon occurs when the growth and survival of individuals within a population are negatively affected by the presence of nearby conspecifics (individuals of the same species) [41]. In the case of the dominant species *P. laurifolia*, *P. alba*, *S. alba*, and *B. pendula* in the Irtysh River basin, negative density dependence likely contributes to the observed clustered distribution by limiting the expansion of individual clumps and promoting the formation of distinct patches.

Elevation and slope, as topographical factors, can also significantly influence the aggregation intensity of plant species. Variations in elevation can lead to differences in climate, soil moisture, and nutrient availability, all of which can affect plant growth and distribution [42]. Similarly, slope can impact soil stability, erosion rates, and water flow patterns, thereby influencing the suitability of habitats for particular species [43,44]. In the Irtysh River basin, these topographical factors may have contributed to the clustered distribution of the dominant species by creating favorable microhabitats for their growth and reproduction.

Finally, the distance from the river channel is another important factor that can affect the aggregation intensity of plant populations. As mentioned earlier, the availability of water and nutrients near the river is crucial for plant growth [45]. Therefore, species that are more tolerant to wet conditions, such as *S. alba*, tend to aggregate closer to the river to take advantage of these resources. In contrast, species that can tolerate drier conditions, like *P. alba*, may exhibit a more dispersed distribution further away from the river.

Overall, the aggregation intensity of the dominant species in the Irtysh River basin is a complex interplay between negative density dependence, elevation, slope, distance from the river channel, and other ecological and biological factors. Understanding these relationships is essential for developing effective conservation and management strategies for the valley forests in this region.

## 5. Conclusions

In conclusion, the dominant species in the plain valley forests of the Irtysh River basin exhibited weak regeneration and a clustered distribution pattern. Factors such as a negative density dependence, elevation, slope, and distance from the river channel significantly influenced the aggregation intensities of these species. To address insufficient regeneration, a series of protection and management measures should be implemented. For example, establishing protected areas, restricting logging and grazing activities, and restoring natural hydrological conditions may prove beneficial. Additionally, these ecological and environmental factors jointly influence the spatial clustering patterns of the dominant species through various mechanisms, revealing their adaptability to their growth environments. Therefore, in the cultivation and management of plain valley forests in the Irtysh River basin, it is essential to maintain the clustered growth characteristics of the populations to provide favorable habitats for their growth and development.

**Author Contributions:** J.S., Z.X. and B.Y. contributed equally to this paper, and they are joint first authors. Conceptualization, J.S. and T.L.; methodology, J.S., Z.X., B.Y. and T.L.; software, J.S.;

validation, J.S. and Z.X.; formal analysis, J.S. and B.Y.; investigation, Y.Y., L.X. and Z.Z.; resources, Y.Y. and L.X.; data curation, Z.Z.; writing—original draft preparation, J.S., Z.X. and B.Y.; writing—review and editing, J.S., Z.X., B.Y. and T.L.; visualization, J.S., Z.X. and B.Y.; supervision, T.L.; project administration, T.L.; funding acquisition, T.L. All authors have read and agreed to the published version of the manuscript.

**Funding:** This research was supported by the Third Xinjiang Scientific Expedition Program, grant number 2021xjkk0603. Founder: Ministry of Science and Technology.

**Data Availability Statement:** The raw data supporting the conclusions of this article will be made available by the authors without undue reservation.

**Conflicts of Interest:** All authors were employed by Xinjiang Production and Construction Corps Key Laboratory of Oasis Town and Mountain Basin System Ecology.

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
