# Peer review of "Spatial Distribution Patterns and Influencing Factors of Dominant Species in Plain Valley Forests of the Irtysh River Basin"

_forests, doi:10.3390/f15071237_

Round 1

Reviewer 1 Report

Comments and Suggestions for Authors

The manuscript of Song et al brings insights into the Spatial distribution patterns and influencing factors of dominant species in plain valley forests of the Irtysh River basin. I found that the ms is well-written. The discussion section included several direct statements that are neither supported by the analysis nor evidence from the earlier studies. Considering the current form, I have the following concerns:

General comments

·         Most of the references used in the introduction are from 2015 ́s or before. Please, update as many as possible.

·         Authors are required to enhance the keywords relevant to the study.

·         Rational behind your objectives are not clearly mentioned in the introduction section

·         Cite recent works in the introduction section.

·         Suggest reshaping the last paragraph of the introduction to state the aims and hypothesis of the study clearly.

·         In material and method, author have to clearly explain how four plant communities

·         categorised more clearly.

·         Add a description of the soil characteristics and dominates trees species in the methodology.

·         Provide clearly the phytosociology (details about composition of tree, shrubs and herbs

·         species)

·         Mention any statistical significance (p-values) where relevant to emphasize the robustness

·         of the findings.

·          Discussion section need to be revised with relevant to the ongoing study in this field.

·         Authors need to emphasize on presented the key/important results and discuss the trend and significance of those results in the discussion section.

·         Expand the discussion to interpret the results in the context of ecological theory and

·         previous studies

·         The English should be improved

Comments on the Quality of English Language

The manuscript of Song et al brings insights into the Spatial distribution patterns and influencing factors of dominant species in plain valley forests of the Irtysh River basin. I found that the ms is well written. The discussion section included several direct statements that are neither supported by the analysis nor evidences from the earlier studies. Considering the current form, I have following concerns:

General comments

·         Most of the references used in the introduction are from 2015 ́s or before. Please, update as many as possible.

·         Authors are required to enhance the keywords relevant to the study.

·         Rational behind your objectives are not clearly mentioned in the introduction section

·         Cite recent works in the introduction section.

·         Suggest reshaping the last paragraph of the introduction to state the aims and hypothesis of the study clearly.

·         In material and method, author have to clearly explain how four plant communities

·         categorised more clearly.

·         Add a description of the soil characteristics and dominates trees species in the methodology.

·         Provide clearly the phytosociology (details about composition of tree, shrubs and herbs

·         species)

·         Mention any statistical significance (p-values) where relevant to emphasize the robustness

·         of the findings.

·          Discussion section need to be revised with relevant to the ongoing study in this field.

·         Authors need to emphasize on presented the key/important results and discuss the trend and significance of those results in the discussion section.

·         Expand the discussion to interpret the results in the context of ecological theory and

·         previous studies

·         The English should be improved

Author Response

Comments 1: Most of the references used in the introduction are from 2015 ́s or before. Please, update as many as possible.

Response 1: Thank you for pointing this out. We agree with this comment. Therefore, we have updated some of the earlier published references in the introduction, mainly as follows:

Reference [9] on line 43, reference [11,12] on line 46, reference [14] on line 50, reference [15] on line 51, and reference [21] on line 66.

9. Havrdová, A., Douda, J., Doudová, J. Threats, biodiversity drivers and restoration in temperate floodplain forests related to spatial scales. Science of the Total Environment, 2023, 854, 158743.

11. Madeira, D.M., Matzner, R.D.S., Filipe Pereira de Lima e Silva, V., Barcelos, G., de Souza, C.R., Rodrigues Santos, L., Araújo, F.D.C., Manoel dos Santos, R. Flooding drives tropical dry forest tree community assembly in Southeast Brazil. Nordic Journal of Botany, 2023, e03913.

12. Herrera-Bevan, E.J., Ibáñez, I. Detrimental impacts of flooding conditions on native tree recruitment but not on invasive plants. Forest Ecology and Management, 2024, 561, 121886.

14. Tauqeer, H.M., Turan, V., Farhad, M., Iqbal, M. Sustainable agriculture and plant production by virtue of biochar in the Era of climate change. In: Hasanuzzaman, M., Ahammed, G.J., Nahar, K. (eds) Managing Plant Production Under Changing Environment. Springer, Singapore, 2022, pp. 21-42.

15. Li Y, Wei L, Ye S. Proportion and distribution of structural frameworks in natural forests. Global Ecology and Conservation, 2023, 48, e02720.

21. Xue, Z.F., Liu, T., Wang, L.S., Song, J.H., Chen, H.Y., Xu, L., Yuan, Y. Community structure and characteristics of plain valley forests in main tributaries of Ertix River Basin, China. Chinese Journal of Plant Ecology, 2024, 48, 390-402.

Comments 2: Authors are required to enhance the keywords relevant to the study.

Response 2: We sincerely appreciate the valuable comment. We agree with this comment. Therefore, we have changed the keyword to “Population distribution pattern 1; Negative density dependence 2; Environmental adaptability 3; Diameter class structure 4; Plain valley forests 5” Line 37-38.

Comments 3: Suggest reshaping the last paragraph of the introduction to state the aims and hypothesis of the study clearly.

Response 3: Thank you for your insightful feedback. Regarding your suggestion to reshape the last paragraph of the introduction to clearly state the aims and hypothesis of the study, we have carefully considered your point.

Indeed, our three research objectives serve as the cornerstone of our study's purpose, which has been articulated in the introduction. However, regarding the hypothesis, we have taken a more exploratory approach in this study, aiming to uncover new insights and trends through data analysis and interpretation, rather than simply validating or refuting predefined assumptions. As such, we chose to focus the introduction on posing questions and outlining our research content, reserving specific hypotheses and anticipated outcomes for the discussion section, where they can be contextualized within the findings.

We understand the value of clearly stating hypotheses in some research contexts, but in this case, we believe our approach aligns with the exploratory nature of our investigation. Nevertheless, we are open to further refining the balance between stating research aims and hinting at potential outcomes in the introduction, should you have specific recommendations on how to achieve this.

Please feel free to provide any further feedback or requests for modification during the subsequent review process. We are grateful for your patience and guidance, and we will work diligently to ensure the clarity and impact of our study.

Comments 4: In material and method, author have to clearly explain how four plant communities categorised more clearly.

Response 4: Thank you very much for your thorough review and valuable comments on our manuscript. We appreciate your attention to our work and understand your concern regarding the categorization of plant communities. However, we would like to clarify that our study does not involve the classification of four plant communities as you have mentioned. Instead, it focuses solely on analyzing the distribution patterns of dominant species within a river valley forest ecosystem.

In the Materials and Methods section, we have detailed how we determined the dominant species in the river valley forest and subsequently analyzed their spatial distribution patterns. Specifically, we have described the following steps:

1.      Data Collection: Through comprehensive field surveys and sample collections, we gathered data on the distribution of plant species within the river valley forest, including species identity, abundance, and growth status.

2.      Identification of Dominant Species: Using the Importance Value (IV) as a quantitative metric, we calculated the relative importance of each plant species in the community and identified the dominant species. These dominant species formed the focus of our analysis.

3.      Distribution Pattern Analysis: For the identified dominant species, we further analyzed their spatial distribution patterns.

We understand that there may have been some confusion regarding the scope of our study, and we apologize for any misunderstanding. To address this, we are willing to revise the relevant sections of the manuscript to make it clearer that our work does not involve the classification of plant communities but rather focuses on the distribution patterns of dominant species within a specific ecosystem.

We sincerely appreciate your feedback and look forward to your further guidance and suggestions to improve the quality of our research.

Comments 5: Add a description of the soil characteristics and dominates trees species in the methodology.

Response 5: Thank you for your review and your thoughtful input regarding the addition of soil characteristics and dominant tree species descriptions in the methodology section of our manuscript.

We appreciate your suggestion, and we understand your concern about the comprehensiveness of our study. However, we would like to clarify that our research did not involve a detailed analysis of soil characteristics. Our primary focus was on the distribution patterns of dominant tree species within the river valley forest ecosystem, and as such, we did not collect or analyze soil samples as part of this study.

Regarding the tree species, we have already included a description of the major tree species in the study area in the relevant section study area section of our manuscript.

We believe that this approach is appropriate for our study, as it allows us to focus on the research questions we aimed to address without detracting from the core objectives of our work. We hope that this clarification addresses your concern, and we remain open to any further suggestions you may have to improve the clarity and comprehensiveness of our manuscript.

Comments 6: Provide clearly the phytosociology (details about composition of tree, shrubs and herbs species).

Response 6: Thank you for your continued review and insightful comments regarding the inclusion of phytosociological details in our manuscript. We understand your request for a more comprehensive description of the vegetation community, including the composition of trees, shrubs, and herb species.

Regarding your suggestion, we acknowledge that a complete phytosociological assessment of the study area would indeed provide a richer understanding of the ecosystem. However, we would like to clarify that our study was specifically designed to focus on the dominant tree species within the river valley forest ecosystem. This focus was driven by our research questions and objectives, which centered around the distribution patterns and potential ecological drivers of these tree species.

While we appreciate the value of a more holistic approach, including shrubs and herbaceous species in our analysis would significantly broaden the scope of our study and potentially detract from our primary objectives. Additionally, collecting and analyzing data on shrubs and herbs would require additional resources and time, which were not available within the constraints of our project.

To address your concern, we suggest that we emphasize in the manuscript the specific focus of our study and clarify that our analysis is limited to dominant tree species. We can also acknowledge the potential importance of shrubs and herbaceous species in the ecosystem and suggest that future studies may benefit from a more comprehensive phytosociological assessment.

We hope that this approach will satisfy your request for clarity while also respecting the limitations of our study design. Please let us know if you have any further suggestions or concerns, and we will be happy to discuss them further.

Comments 7: Mention any statistical significance (p-values) where relevant to emphasize the robustness of the findings.

Response 7: Thank you for your thoughtful review and for highlighting the importance of mentioning statistical significance (p-values) in our manuscript. Your observation is well-taken, and I apologize for the oversight in not clearly presenting this critical aspect of our analysis.

As you suggested, we have now revised the manuscript to include relevant p-values wherever statistical significance is appropriate. These additions help emphasize the robustness of our findings and provide readers with a clearer understanding of the statistical support behind our conclusions. – Line 311-313.

We believe that these revisions significantly strengthen our manuscript by providing a more complete and transparent picture of our analytical approach and the statistical support for our findings.

Once again, thank you for your insightful feedback. We appreciate your time and effort in reviewing our work and helping us improve the quality of our manuscript.

Comments 8: Discussion section need to be revised with relevant to the ongoing study in this field. Authors need to emphasize on presented the key/important results and discuss the trend and significance of those results in the discussion section. Expand the discussion to interpret the results in the context of ecological theory and previous studies.

Response 8: I would like to express my sincere gratitude for your thoughtful review of our manuscript and the valuable insights you have provided, particularly regarding the discussion section. Your comments have been instrumental in guiding us towards enhancing the quality and depth of our work.

In response to your suggestions, we have made the following revisions:

“Negative density dependence, as a manifestation of population competition, plays a crucial role in shaping the spatial distribution of plant populations [40]. This phenomenon occurs when the growth and survival of individuals within a population are negatively affected by the presence of nearby conspecifics (individuals of the same species) [41]. In the case of the dominant species P. laurifolia, P. alba, S. alba, and B. pendula in the Irtysh River basin, negative density dependence likely contributes to the observed clustered distribution by limiting the expansion of individual clumps and promoting the formation of distinct patches.

Elevation and slope, as topographical factors, can also significantly influence the aggregation intensity of plant species. Variations in elevation can lead to differences in climate, soil moisture, and nutrient availability, all of which can affect plant growth and distribution [42]. Similarly, slope can impact soil stability, erosion rates, and water flow patterns, thereby influencing the suitability of habitats for particular species [43,44]. In the Irtysh River basin, these topographical factors may have contributed to the clustered distribution of the dominant species by creating favorable microhabitats for their growth and reproduction.

Finally, the distance from the river channel is another important factor that can affect the aggregation intensity of plant populations. As mentioned earlier, the availability of water and nutrients near the river is crucial for plant growth [45]. Therefore, species that are more tolerant to wet conditions, such as S. alba, tend to aggregate closer to the river to take advantage of these resources. In contrast, species that can tolerate drier conditions, like P. alba, may exhibit a more dispersed distribution further away from the river.

Overall, the aggregation intensity of the dominant species in the Irtysh River basin is a complex interplay between negative density dependence, elevation, slope, distance from the river channel, and other ecological and biological factors. Understanding these relationships is essential for developing effective conservation and management strategies for the valley forests in this region.” – Line 392-418.

Please let us know if you have any further comments or if any other clarifications are needed. We are committed to ensuring that our manuscript meets the highest standards of quality and academic rigor.

Reviewer 2 Report

Comments and Suggestions for Authors

General Comments:

  • The manuscript provides valuable insights into the spatial distribution and influencing factors of dominant species in the Irtysh River basin.
  • The study is well-structured but requires improvements in clarity, grammar, and scientific rigor in certain sections.

Abstract

·       Line 16-17: Rephrase for clarity: "Previous studies mainly focused on individual tributaries or main streams, lacking comprehensive research on the overall river and valley forest resources and their ecological functions."

·       Line 19-20: Define "comprehensive investigations" more clearly.

·       Line 27: Instead of "aggregation intensity," consider using "density" or "concentration" consistently.

·       Line 31: Specify the negative effect on "aggregation intensities" for clearer interpretation.

Introduction

·       Line 39: Cite relevant sources accurately.

·       Line 41-43: Rephrase for readability: "Valley forests accumulate carbon faster than other dryland forests, contributing significantly to rapid carbon sequestration."

·       Line 46-48: Clarify the roles of these forests in ecosystem integrity and climate change mitigation.

·       Line 49-51: Simplify complex sentences for better readability.

·       Line 63-64: Clarify ecological functions by providing specific examples.

·       Line 71-73: Focus on the necessity of comprehensive documentation by specifying what has been lacking in previous studies.

Methods;

·       Line 101: Repeated content "valley forests" which can be confusing.

·       Lines 106-107: The temperature data lacks units in the text.

·       Lines 195; The term "dbh99\2" and "dbh99\1" are confusing and not standard notation.

·       Lines 218-223: Repetitive wording and lack of clarity in describing abiotic factors.

·       Line 239: Mention of software without specifying versions or contexts.

Results

·       Lines 250-254: Redundant wording and lack of clarity.

·       Lines 279-287: Repetitive wording and lack of clarity.

Discussion

·       Line 322:  Other species of the Salicaceae family were less abundant and sparsely distributed in a mosaic pattern within the valley forest community. Clarify which species are being referred to and ensure the sentence fits smoothly into the context of the paragraph.

Author Response

Comments 1: Line 16-17: Rephrase for clarity: "Previous studies mainly focused on individual tributaries or main streams, lacking comprehensive research on the overall river and valley forest resources and their ecological functions."

Response 1: Thank you for revising the text to enhance its readability. We appreciate the changes you have made, which now reads: "Previous studies mainly focused on individual tributaries or main streams, lacking comprehensive research on the overall river and valley forest resources and their ecological functions." - Line 14-16.

Comments 2: Line 19-20: Define "comprehensive investigations" more clearly.

Response 2: Thank you for your valuable feedback on our manuscript. In response to your request to define "comprehensive investigations" more clearly, our explanation is mainly as follows:

The habitat determines plant community structure (Li et al., 2024). We conducted a comprehensive reconnaissance survey of the overall distribution of valley forests, with a focus on identifying the different habitat types that arise with elevation gradients, as well as areas of dominant species distribution. Therefore, the distance between sample plots varied between 6 and 8 kilometers.

Furthermore, within each of these predefined intervals, we carefully selected four replicate plots that were deemed representative of the dominant species composition and spatial distribution patterns in that particular area. This approach allowed us to capture the intricate variations and nuances of the valley forests across various branches of the Irtysh River basin plain.

Thus, our 'comprehensive investigations' refer to this rigorous, systematic, and representative sampling methodology, which aims to comprehensively characterize and understand the valley forests of the Irtysh River basin plain, addressing the research gap in a comprehensive and nuanced manner.

We hope this revised section clarifies our methodology and meets your expectations. If you have any further suggestions or require any additional information, please do not hesitate to let us know. Thank you again for your time and consideration.

Li, X.; Li, X.; Zhang, M.; Luo, Q; Li, Y.; Dong, L. Urban park attributes as predictors for the diversity and composition of spontaneous plants− A case in Beijing, China. URBAN FOR URBAN GREE, 2024, 91, 128185.

Comments 3: Line 27: Instead of "aggregation intensity," consider using "density" or "concentration" consistently.

Response 3: We appreciate the reviewer's suggestion, but we believe "aggregation intensity" more accurately conveys the concept we are discussing. While "density" or "concentration" refer to the number of elements per unit area or volume, "aggregation intensity" specifically captures the degree to which elements are clustered together, reflecting both their spatial arrangement and their frequency. This term is essential to describe the observed patterns and implications in our study, which focus on the intensity of clustering rather than just numerical density.

31.   Chen, P., Xia, J.B., Ma H.S., Gao F.L., Dong M.M., Xing X.S., Li, C.R. Analysis of spatial distribution pattern and its influencing factors of the Tamarix chinensis population on the beach of the muddy coastal zone of Bohai Bay. Ecological Indicators, 2022, 140, 109016.

Comments 4: Line 31: Specify the negative effect on "aggregation intensities" for clearer interpretation.

Response 4: Thank you for your insightful comment on our manuscript. To clarify the negative effects on "aggregation intensities" and provide a more detailed interpretation, we have revised the relevant sentence as follows:

"Among the abiotic factors, elevation had a significant positive effect on the aggregation intensities of P. alba, S. alba, and B. pendula, indicating that these species tend to aggregate more densely at higher elevations. Conversely, slope had a significant negative impact on the aggregation intensities of P. laurifolia, P. alba, and S. alba, suggesting that increasing slope steepness leads to a decrease in the clustering of these species. Similarly, the distance from the river channel had a significant negative effect on the aggregation intensities of S. alba and B. pendula, implying that as the distance from the river increases, the clustering patterns of these species become less pronounced." – Line 28-35.

We hope this revised sentence better explains the negative effects on aggregation intensities and provides a clearer interpretation of our results. If you have any further suggestions or require any additional information, please do not hesitate to let us know. Thank you again for your time and consideration.

Comments 5: Line 39: Cite relevant sources accurately.

Response 5: We appreciate your guidance in ensuring that our references adhere to the journal's requirements.

In response to your comment regarding citing relevant sources accurately, we have carefully reviewed and revised our citations accordingly. For the specific case of the "A Guide to Bottomland Hardwood Restoration" by Allen et al., which is a technical report rather than a journal article, we have adjusted the citation to comply with the journal's preferred format for non-journal publications.

Here is the revised citation:

“Allen, J.A., Keeland, B.D., Stanturf, J.A., Clewell, A.F., Harvey, E., Kennedy, Jr. A Guide to Bottomland Hardwood Restoration. Information and Technology Report USGS/BRD/ITR–2000-0011. U.S. Geological Survey, Biological Resources Division. Asheville, NC: U.S. Department of Agriculture, Forest Service, Southern Research Station. General Technical Report SRS–40. 2001 (Revised 2004). DOI: https://doi.org/10.2737/SRS-GTR-40.”

Please note that for non-journal publications such as this technical report, the standard convention does not involve a "journal name" field. Instead, we have included the report number, publishing organization, and the date of publication (or revision, if applicable). The year has been placed after the relevant publishing information, as requested by the journal.

We have carefully reviewed all other citations in our manuscript to ensure that they also conform to the journal's guidelines. If you find any further discrepancies or have any additional suggestions, please do not hesitate to let us know.

Thank you again for your valuable feedback. We look forward to your continued support and guidance as we work to improve our manuscript.

Comments 6: Line 41-43: Rephrase for readability: "Valley forests accumulate carbon faster than other dryland forests, contributing significantly to rapid carbon sequestration."

Response 6: Thank you for revising the text to enhance its readability. We appreciate the changes you have made, which now reads: " Valley forests accumulate carbon faster than other dryland forests, contributing significantly to rapid carbon sequestration." - Line 44-45.

Comments 7: Line 46-48: Clarify the roles of these forests in ecosystem integrity and climate change mitigation.

Response 7: Thank you for your thoughtful comments and valuable suggestions. We appreciate your emphasis on clarifying the roles of valley forests in ecosystem integrity and climate change mitigation.

Indeed, valley forests are essential components of the natural environment, playing multifaceted roles in maintaining ecosystem health and resilience. They provide vital habitats for diverse biodiversity, fostering the coexistence of numerous species and supporting complex food webs. By regulating water quality and quantity, these forests help to maintain the balance of aquatic ecosystems downstream, ensuring clean and abundant water resources for both humans and wildlife. Furthermore, valley forests contribute to nutrient cycling, which is crucial for soil fertility and plant growth, sustaining the productivity of terrestrial ecosystems.

In addition to their ecological functions, valley forests play a significant role in mitigating climate change. They sequester and store substantial amounts of carbon, reducing atmospheric greenhouse gas concentrations and contributing to global carbon storage efforts. This carbon sequestration helps to regulate local climates, moderating temperature extremes and protecting against the adverse effects of climate change.

We believe that these clarifications address your concerns and provide a more comprehensive understanding of the importance of valley forests in maintaining ecosystem integrity and mitigating climate change. If you have any further suggestions or require any additional information, please do not hesitate to contact us.

Comments 8: Line 49-51: Simplify complex sentences for better readability.

Response 8: Thank you for your valuable suggestions. We have revised "The spatial pattern of plant populations refers to their typical spatial distribution structure, which depends on the ecological niches of plant resources, plant competition, and environmental adaptability" to "The typical spatial distribution structure of plant populations is determined by their ecological niches, competition among plants, and environmental adaptability." - Line 51-53.

Comments 9: Line 63-64: Clarify ecological functions by providing specific examples.

Response 9: Thank you for your valuable suggestion to clarify ecological functions through specific examples. While we understand the importance of illustrating these functions with concrete cases, we have opted to maintain the brevity and generality of our discussion in order to avoid overwhelming the reader with details that may be specific to individual ecosystems. However, we acknowledge that specific examples can enrich understanding and invite readers to consult related literature, such as case studies on riverine ecosystems and forest functions, for a more in-depth exploration of these topics. If you have any particular examples in mind that you believe would strengthen our argument, we would be happy to consider incorporating them into a future revision.

Comments 10: Line 71-73: Focus on the necessity of comprehensive documentation by specifying what has been lacking in previous studies.

Response 10: Thank you very much for your thoughtful review of our manuscript and for your insightful comments. We have carefully considered your suggestions, particularly your emphasis on the necessity of comprehensive documentation and the need to specify what has been lacking in previous studies.

Indeed, as you pointed out, previous research on river and valley ecosystems, though extensive in areas such as poplar genetic diversity, ecological water demand, ecosystem assessments, plant diversity, and vegetation changes, has often been confined to individual tributaries or main streams within the study basin. This has resulted in a fragmented understanding of the broader context and interconnectedness of valley forests across the entire system. Notably, the distribution and ecological roles of dominant species within these forests have received limited attention, which is crucial for assessing their resilience and sustainability.

To address this gap, we have revised the manuscript to emphasize the pressing need for a comprehensive documentation of the current state of river and valley forest resources, including a detailed assessment of their ecological functions and the distribution of dominant species. We believe that such a comprehensive approach will provide a more holistic view of these valuable ecosystems and serve as a foundation for effective conservation, management, and restoration efforts.

We are grateful for your guidance and for highlighting the importance of this aspect of our work. If you have any further suggestions or concerns, please do not hesitate to let us know. We are committed to ensuring that our manuscript meets the highest standards of scientific rigor and clarity.

“Although some scholars have studied poplar genetic diversity [23,24], ecological water demand and dynamics of forests and grasslands in valleys [25], ecosystem assessments [26,27], plant diversity [28], and vegetation changes [29], a significant limitation of these studies lies in their narrow focus. Specifically, most prior works have been constrained to individual tributaries or the main streams within the study basin, failing to capture the broader context and interconnectedness of valley forests across the entire valley system. Furthermore, little attention has been paid to the distribution and ecological roles of dominant species within these forests, which are vital for understanding their resilience and sustainability. Therefore, there is a pressing need for a comprehensive documentation of the current state of river and valley forest resources, including a detailed assessment of their ecological functions and the distribution of their dominant species. Such an endeavor would fill the gaps in our knowledge and provide a solid foundation for the effective conservation, management, and restoration of these valuable ecosystems.” – Line 73-86.

Comments 11: Line 101: Repeated content "valley forests" which can be confusing.

Response 11: Thank you for your valuable feedback regarding the repeated use of the term "valley forests" in our manuscript. We understand that it may have caused confusion, and we apologize for any inconvenience this may have caused.

As you noted, the Irtysh River basin includes the main stream of the Irtysh River as well as its tributaries, namely the Kara Irtysh, Crane, Burqin, Haba, and Berezek Rivers. Each of these rivers hosts valley forests in their respective valleys, which is why the term "valley forests" appears repeatedly in our description.

To address this issue, we will revise the text to reduce repetition and clarify the distribution of valley forests in the basin. One possible approach could be to rewrite the sentence as follows:

"The Irtysh River basin encompasses the main stream of the Irtysh River and its primary tributaries, including the Kara Irtysh, Crane, Burqin, Haba, and Berezek Rivers. Each of these rivers supports a unique distribution of valley forests along their respective valleys, contributing to the diverse and rich ecosystem of the entire basin." – Line 108-111.

We welcome any further suggestions you may have to improve the clarity and conciseness of our manuscript. Please feel free to share your thoughts with us, and we will incorporate them into our revisions accordingly.

Comments 12: Lines 106-107: The temperature data lacks units in the text.

Response 12: Thank you for pointing out the oversight in the temperature data presented in our manuscript. You are correct that the units for the temperature measurements were omitted, which can lead to confusion.

To rectify this, we will revise the sentence to include the appropriate units for both the average annual temperature and the average annual minimum temperature. The revised sentence will read as follows:

"The average annual temperature is 4.2°C, with the average annual minimum temperature reaching -28.1°C." – Line 116-117.

We apologize for any inconvenience caused by this oversight and appreciate your careful review of our work. Your input is invaluable in ensuring the clarity and accuracy of our manuscript.

Comments 13: Lines 195: The term "dbh99\2" and "dbh99\1" are confusing and not standard notation.

Response 13: Thank you for bringing up the confusion regarding the notation "dbh99\2" and "dbh99\1" in our manuscript. We fully understand your concern that these notations may not be immediately clear to readers outside of the specific context where they were originally used.

As you mentioned, these notations are indeed derived from previous works by Bagchi et al. (2011) and Lu et al. (2021), which we have cited in our manuscript. Our intention in using these notations was to maintain consistency with the original studies and ensure that our methods are transparent and reproducible.

However, we acknowledge that the lack of standard notation for these calculations may cause confusion for some readers. To address this issue, we propose the following revisions:

Clarify the notation: We will add a brief explanation of the notation immediately after its first use, clarifying that "dbh992/3" refers to the power of two-thirds of the 99th percentile DBH, and "dbh991/2" refers to the power of half of the 99th percentile DBH. – Line 212-214.

Consider alternative notation: If the explanation still proves confusing, we will explore alternative notations that are more intuitive and widely recognized. For example, we could use "sqrt(dbh99)" and "dbh99^(1/2)" to represent the square root of the 99th percentile DBH, and clarify that trees with DBH greater than or equal to this value are classified as adults.

Check for typos: We will carefully review the manuscript to ensure that there are no typos or inconsistencies in the notation, particularly regarding "dbh99\1", which may have been a misinterpretation or transcription error.

We would appreciate your guidance on whether these proposed revisions are sufficient to address your concerns. If you have any specific suggestions for alternative notations or explanations, please do not hesitate to share them with us.

32.   Bagchi, R., Henrys, P.A., Brown, P.E., Burslem, D.F.P., Diggle, P.J., Gunatilleke, C.S., Gunatilleke, I.U.N., Kassim, A.R., Law, R., Noor, S., Valencia, R.L. Spatial patterns reveal negative density dependence and habitat associations in tropical trees. Ecology, 2011, 92, 1723-1729.

33.   Lu, M.Z., Du, H., Song, T.Q., Peng, W.X., Su, L., Zhang, H., Zeng, Z.X., Wang, K.L., Zeng, F.P. Effects of density dependence in an evergreen-deciduous broadleaf karst forest in southwest China. Forest Ecology and Management, 2021, 490, 119142.

Comments 14: Lines 218-223: Repetitive wording and lack of clarity in describing abiotic factors.

Response 14: Thank you for your valuable suggestions. We have revised this paragraph to read: "Abiotic factors such as climate, topography, soil conditions, hydrological characteristics, and natural disasters collectively influence the distribution and growth of plants. However, this study primarily focuses on the impact of topography on the spatial distribution of dominant species in river valley forests. Topographical features, including elevation, slope, and distance from the river channel, affect the water and nutrient availability for plant growth." – Line 239-244.

Please let us know if you have any further comments or if any other clarifications are needed. We are committed to ensuring that our manuscript meets the highest standards of quality and academic rigor.

Comments 15: Line 239: Mention of software without specifying versions or contexts.

Response 15: Thank you for your careful review and suggestions. We have added the software versions to the data processing section as follows: “R version 4.3.3 was used to perform the study’s analyses, with the MASS and stats packages employed for negative density dependence intensity analysis, the lavaan package for structural equation model construction, and the vegan package for the variance decomposition analysis. Data were collated using Excel 2021, Origin 2021, and ArcGIS Pro 3.0.2 software were used for plotting.” – 260-264.

Please let us know if you have any further comments or if any other clarifications are needed. We are committed to ensuring that our manuscript meets the highest standards of quality and academic rigor.

Comments 16: Lines 250-254: Redundant wording and lack of clarity.

Response 16: Thank you for pointing this out. We agree with this comment. Therefore, we have revised it to: “An analysis of the tree species composition and important values in the plain valley forests of various tributaries and the mainstream of the Irtysh River basin revealed that there are a total of 10 tree species, among which P. laurifolia, P. alba, S. alba, and B. pendula emerged as the dominant species in the region." – Line 270-273.

Please let us know if you have any further comments or if any other clarifications are needed. We are committed to ensuring that our manuscript meets the highest standards of quality and academic rigor.

Comments 17: Lines 279-287: Repetitive wording and lack of clarity.

Response 17: Thank you for pointing this out. We agree with this comment. Therefore, we have revised it to: " In the plain valley forests along the Irtysh River basin, significant differences in aggregation intensity were observed among the P. laurifolia, P. alba, S. alba, and B. pendula populations in various tributaries and mainstream plain valley forests, indicating their distinct biological characteristics and varying abilities to adapt to the environment. The lower aggregation intensity of P. laurifolia, P. alba, and S. alba in the mainstream of the Irtysh River compared to its tributaries suggests that the mainstream environment is more conducive to their growth." – Line 298-304.

Please let us know if you have any further comments or if any other clarifications are needed. We are committed to ensuring that our manuscript meets the highest standards of quality and academic rigor.

Comments 18:  Line 322: Other species of the Salicaceae family were less abundant and sparsely distributed in a mosaic pattern within the valley forest community. Clarify which species are being referred to and ensure the sentence fits smoothly into the context of the paragraph.

Response 18: Thank you for pointing this out. We agree with this comment. Therefore, we have revised it to: "Other species of the Salicaceae family, such as Populus nigra, Populus canescens, Populus euphratica, and Salix burqinensis, were present in smaller numbers and distributed in a scattered and intermixed manner within the valley forest community." – Line 341-344.

Please let us know if you have any further comments or if any other clarifications are needed. We are committed to ensuring that our manuscript meets the highest standards of quality and academic rigor.

Reviewer 3 Report

Comments and Suggestions for Authors

The manuscript is well written and based on high quality field data. Although the acquired data, employed methods and results are interesting and appropriate for answering the questions posed by the present study, I consider the discussion section is too descriptive. Therefore, I think that discussion should be improved, by using the obtained results to explain complex ecological patterns, make inferences regarding forest functions (which are repeatedly mentioned in the introduction section) and suggestions regarding conservation measures if possible.

In addition, I have a minor comment regarding the manuscript, i.e. a caption as well as a reference in the text should be added for the image in line 216.

Comments on the Quality of English Language

I consider the quality of the english language as highly adequate.

Author Response

Comments 1: The manuscript is well written and based on high quality field data. Although the acquired data, employed methods and results are interesting and appropriate for answering the questions posed by the present study, I consider the discussion section is too descriptive. Therefore, I think that discussion should be improved, by using the obtained results to explain complex ecological patterns, make inferences regarding forest functions (which are repeatedly mentioned in the introduction section) and suggestions regarding conservation measures if possible.

Response 1: I would like to express my sincere gratitude for your thoughtful review of our manuscript and the valuable insights you have provided, particularly regarding the discussion section. Your comments have been instrumental in guiding us towards enhancing the quality and depth of our work.

In response to your suggestions, we have made the following revisions:

“Negative density dependence, as a manifestation of population competition, plays a crucial role in shaping the spatial distribution of plant populations [40]. This phenomenon occurs when the growth and survival of individuals within a population are negatively affected by the presence of nearby conspecifics (individuals of the same species) [41]. In the case of the dominant species P. laurifolia, P. alba, S. alba, and B. pendula in the Irtysh River basin, negative density dependence likely contributes to the observed clustered distribution by limiting the expansion of individual clumps and promoting the formation of distinct patches.

Elevation and slope, as topographical factors, can also significantly influence the aggregation intensity of plant species. Variations in elevation can lead to differences in climate, soil moisture, and nutrient availability, all of which can affect plant growth and distribution [42]. Similarly, slope can impact soil stability, erosion rates, and water flow patterns, thereby influencing the suitability of habitats for particular species [43,44]. In the Irtysh River basin, these topographical factors may have contributed to the clustered distribution of the dominant species by creating favorable microhabitats for their growth and reproduction.

Finally, the distance from the river channel is another important factor that can affect the aggregation intensity of plant populations. As mentioned earlier, the availability of water and nutrients near the river is crucial for plant growth [45]. Therefore, species that are more tolerant to wet conditions, such as S. alba, tend to aggregate closer to the river to take advantage of these resources. In contrast, species that can tolerate drier conditions, like P. alba, may exhibit a more dispersed distribution further away from the river.

Overall, the aggregation intensity of the dominant species in the Irtysh River basin is a complex interplay between negative density dependence, elevation, slope, distance from the river channel, and other ecological and biological factors. Understanding these relationships is essential for developing effective conservation and management strategies for the valley forests in this region.” – Line 392-418.

Please let us know if you have any further comments or if any other clarifications are needed. We are committed to ensuring that our manuscript meets the highest standards of quality and academic rigor.

Comments 2: I have a minor comment regarding the manuscript, i.e. a caption as well as a reference in the text should be added for the image in line 216.

Response 2: Thank you very much for your thorough review of our manuscript and for bringing up this important oversight. We deeply apologize for not including a caption and a reference in the text for the image mentioned in line 216. This was indeed a careless mistake on our part.

To address your comment, we have made the following revisions:

Caption Addition: We have added a clear and descriptive caption directly below the image to accurately convey its content and purpose. This will enhance the reader's understanding of the image.

Text Reference: In the text, specifically around line 228 where the image is mentioned, we have inserted an appropriate reference marker (e.g., "[Figure 1]") and have included the corresponding entry in the list of figures at the end of the manuscript.

We believe that these modifications will significantly improve the clarity and completeness of our manuscript. We sincerely appreciate your time and effort in reviewing our work and providing valuable feedback.

Please let us know if you have any further comments or if any other clarifications are needed. We are committed to ensuring that our manuscript meets the highest standards of quality and academic rigor.

Round 2

Reviewer 3 Report

Comments and Suggestions for Authors

I consider the response and manuscript alterations adequate for the overall improvement of the manuscript.

Comments on the Quality of English Language

I  consider the quality of the english language as highly adequate.